# Cancer cell-derived interleukin-33 decoy receptor sST2 enhances orthotopic tumor growth in a murine pancreatic cancer model

**Keizo Takenaga**[1]*, **Miho Akimoto**[2], **Nobuko Koshikawa**[1], **Hiroki Nagase**[1]

1 Laboratory of Cancer Genetics, Chiba Cancer Center Research Institute, Nitona, Chuoh-ku, Chiba, Japan,
2 Department of Biochemistry, Teikyo University School of Medicine, Kaga, Itabashi-ku, Tokyo, Japan

* ktakenaga@chiba-cc.jp

**Data Availability Statement:** All relevant data are within the manuscript and its Supporting Information files.

## Abstract

### Background

Proinflammatory interleukin-33 (IL-33) binds to its receptor ST2L and is involved in inflammation and the malignant behavior of cancer cells. However, the role of IL-33-ST2L and the IL-33 decoy receptor sST2 in the tumor microenvironment of pancreatic cancer is unclear. Because we previously reported that sST2 derived from colon cancer cells profoundly influences malignant tumor growth, we hypothesized that sST2 released from pancreatic cancer cells also modulates IL-33-ST2L signaling in the tumor microenvironment, thereby influencing tumor growth.

### Methods

ST2 (ST2L and sST2) expression in mouse pancreatic cancer Panc02 cells was downregulated by shRNAs. mRNA expression levels of IL-33, ST2, cytokines and chemokines in the cells and tumor tissues were examined using real-time PCR. sST2 secretion and the amount of CXCL3 in tumor tissues were measured using ELISA. Tumor growth was investigated after injection of the cells into the pancreas of C57BL/6 mice. MPO[+], F4/80[+] and CD20[+] cells in tumor tissues were detected using immunohistochemistry.

### Results

Some but not all human and mouse pancreatic cancer cell lines preferentially expressed sST2. Then, we investigated the role of sST2 in orthotopic tumor growth of sST2-expressing mouse pancreatic cancer Panc02 cells in immunocompetent mice. shRNA-mediated knockdown of sST2 expression in the cells suppressed orthotopic tumor growth, which was partially recovered by overexpression of shRNA-resistant sST2 mRNA but was not evident in IL-33 knockout mice. This was associated with decreases in *Cxcl3* expression, vessel density and accumulation of cancer-associated neutrophils but not cancer-associated macrophages. Administration of SB225002, an inhibitor of the CXCL3 receptor CXCR2, induced similar effects.

**Funding:** This work was supported in part by JSPS KAKENHI (https://www.jsps.go.jp/) Grant Number 25430110 and by JSPS KAKENHI Grant Number 24791422 to M.A. The funding bodies had no role in the design of the study and data collection, analysis, and interpretation of data and in preparing the manuscript.

**Competing interests:** The authors declare that they have no competing interests.

## Conclusions

Cancer cell-derived sST2 enhances tumor growth through upregulation of CXCL3 via inhibition of IL-33-ST2L signaling in the tumor microenvironment of pancreatic cancer. These results suggest that the sST2 and the CXCL3-CXCR2 axis could be therapeutic targets.

## Introduction

Pancreatic cancer is a disease with a poor prognosis. Most patients already have locally advanced or metastatic disease at the time of diagnosis [1, 2]. Furthermore, pancreatic cancer is very hypoxic and often resistant to radiochemotherapy [3–5]. Therefore, a better understanding of the pathophysiological characteristics of pancreatic cancer is critical for the development of more effective therapeutic approaches for patients with pancreatic cancer.

ST2 is encoded by the *IL1RL1* gene, is a member of the interleukin-1 (IL-1) receptor family [6] and consists of at least two isoforms, ST2L and sST2, which are produced via alternative splicing [7–9]. ST2L is a transmembrane form and is expressed in a variety of cell types, including Th2 lymphocytes, macrophages and NK cells [7–9], whereas sST2 is a soluble form that is predominantly expressed in fibroblasts, epithelial cells and cancer cells [10, 11].

IL-33 has been shown to be primarily expressed as a proinflammatory cytokine by a variety of cell types, such as epithelial cells, myofibroblasts, fibroblasts and macrophages, either constitutively or in response to different stimuli, including chemokines and cytokines [12–14]. IL-33 binds to the cell surface receptor consisting of ST2L and IL-1 receptor accessory protein (IL-1RAP) [15, 16], which is blocked by the decoy receptor sST2 [10, 11].

Recently, the IL-33/ST2L axis has been shown to be involved in the progression of cancer, either positively or negatively, depending on the cancer type, through modulating the tumor microenvironment, such as infiltration of T cells and inflammation. For example, the amount of serum IL-33 is positively correlated with a poor prognosis in gastric cancer [17], non-small cell lung cancer [18], and hepatocellular carcinoma [19]. IL-33 promotes tumor progression in mouse breast, lung and colon cancers [20, 21] and human colon cancer [22]. Conversely, IL-33 suppresses tumor growth and metastasis in mouse melanoma, lung carcinoma and mammary carcinoma [23, 24]. Thus, the effect of IL-33 on tumor progression might be cell type- and context-dependent. With regard to pancreatic cancer, the role of IL-33 largely remains unexplored. On the one hand, IL-33 is implicated as a crucial mediator in inflammation-associated pancreatic carcinogenesis [25], but on the other hand, it induces apoptosis in human MIAPaCa-2 cells [26]. Thus, the role of the IL-33/ST2L axis in regulating pancreatic cancer progression is unresolved.

We previously demonstrated that colon cancer cell-derived sST2 suppresses tumor growth by inhibiting the Th2 response, M2 macrophage polarization and tumor angiogenesis triggered by IL-33 in the tumor microenvironment [22]. To investigate whether sST2 also suppresses tumor growth in pancreatic cancer, we first examined the expression of sST2 in human and mouse pancreatic cancer cell lines. By employing sST2-expressing pancreatic cancer Panc02 cells in an orthotopic implantation mouse model, we report here that, contrary to expectations, sST2 enhanced orthotopic tumor growth in immunocompetent but not IL-33 knockout mice, which suggests that IL-33-ST2L signaling inhibits pancreatic cancer growth.

## Materials and methods

### Reagents

Murine recombinant IL-33 (rIL-33) was purchased from R&D Systems, Inc. (McKinley Place NE, MN, USA). SB225002 was obtained from Selleck Chemicals (Tokyo, Japan).

### Cells and cell culture

Mouse pancreatic cancer Panc02 cells and the human pancreatic cell lines AsPC-1, BxPC3, CFPAC-1, MIAPaCa-2, Panc-1 and SW1990 were used [27]. Panc02 cells were kindly provided by Dr. T. Hollingsworth of the University of Nebraska Medical Center and were characterized elsewhere [27, 28]. Panc-1 and MIAPaCa-2 cells were obtained from the RIKEN BRC Cell Bank (Tsukuba, Japan), and the other human pancreatic cell lines were purchased from ATCC (Manassas, VA, USA). The cells were maintained in Dulbecco's modified Eagle's medium (DMEM) supplemented with 10% fetal bovine serum and 40 μg/ml gentamicin in a humidified atmosphere with 95% air/5% $CO_2$ at 37°C. All cell lines were free of mycoplasma contamination as evaluated using the e-Myco Mycoplasma PCR Detection Kit (Cosmo Bio Co Ltd., Tokyo, Japan).

### ST2 knockdown by shRNAs

For ST2 knockdown in Panc02 cells, MISSION mouse ST2 shRNA lentiviral vectors (#3: TRCN0000058517 for knockdown of both ST2L and sST2 and #5; TRCN0000039054 for knockdown of only sST2) in the pLKO.1-puro plasmid (Sigma-Aldrich Japan, Tokyo, Japan) were used [22]. Lentiviral stocks encoding the shRNA were prepared as previously described [22]. Panc02 cells were transduced with MISSION pLKO.1-puro control transduction particles or shRNA lentiviruses and selected with puromycin (5 μg/ml) to establish control cells (shCont cells) or cells that displayed stable downregulation of both ST2L and sST2 (sh#3 cells) or only sST2 (sh#5 cells), respectively.

### Forced sST2 expression in shsST2 knockdown Panc02 cells

To recover sST2 expression in Panc02-sh#5 cells, the cells were transfected with the plasmid pcDNA3.1/sST2Δ3'-UTR, which lacks the target sequence of #5 ST2 shRNA, using Lipofectamine 2000 according to the manufacturer's protocol. The cells were then selected with 400 μg/ml G418 to establish Panc02-sh#5-sST2 cells. For the control, Panc02-sh#5 cells were transfected with pcDNA3.1 to establish Panc02-sh#5-VC with the same procedure.

### MTT assay

Cell growth and viability were measured by using the MTT (3-(4,5-dimethylthiazol-2-yl)-2,5-diphenyltetrazolium bromide) assay. Briefly, cells ($2 \times 10^4$ cells/well) were cultured in 96-well tissue culture plates and treated in triplicate with different concentrations of rIL-33 or solvent alone for 2 days. At the end of the incubation, 10 μl of MTT (2.5 mg/ml) (Sigma-Aldrich) was added to the wells for 4 h to allow the formation of MTT formazan crystals. After the medium was removed, the crystals were solubilized in 100 μl of DMSO. The absorbance was recorded at 550 nm.

### RNA preparation and qRT-PCR

One microgram of total RNA was extracted from cells or tumor tissues and reverse-transcribed with a ReverTra Ace qPCR RT kit (TOYOBO, Osaka, Japan). qRT-PCR was performed

on cDNA using THUNDERBIRD SYBR qPCR Mix (TOYOBO) and 0.3 μM primers, as shown in S1 Table. The PCR protocol consisted of an initial denaturation step at 95°C for 1 min and 40 cycles of denaturation (95°C for 15 s) and extension (60°C for 1 min). The mRNA expression level of each gene in the cells or tumors was normalized to *Gapdh*.

## Quantification of sST2 and IL-33 by ELISA

Conditioned media were harvested after 24 h of culture and centrifuged at $15,000 \times g$ for 20 min at 4°C. For serum preparation, blood was allowed to clot and centrifuged for 10 min at 1,000 g. sST2 and IL-33 concentrations were measured with the Mouse ST2/IL-1 R4 Quantikine ELISA Kit (R&D Systems) and the Mouse/Rat IL-33 Quantikine ELISA Kit (R&D Systems), respectively, according to the manufacturer's instructions.

## Profiling chemokine and cytokine gene expression

Chemokine and cytokine genes that were differentially expressed in Panc02-shCont versus Panc02-sh#5 cells were profiled by a Mouse Cytokines & Chemokines $RT^2$ Profiler PCR Array (QIAGEN, Hilden, Germany).

## Tumor transplantation

All animal handling and experimental procedures were performed in compliance with the institutional guidelines for the care and use of animals in research. The protocol was approved by the Committee on the Ethics of Animal Experiments of Chiba Cancer Center (Permission Number: 18–1) and the IZUMO Campus Animal Care and Use Committee of Shimane University (Permission Number: IZ26-7). Six- to 7-week-old male C57BL/6J and C57BL/6N mice (20~25 g body weight, CLEA Japan, Osaka, Japan) were used in this study. C57BL/6N-IL-33$^{-/-}$ mice (CDB0631K) [29] were obtained from RIKEN CDB (Kobe, Japan) (http://www.cdb. riken.jp/arg/mutant%20mice%20list.html). The mice were bred in-house and weaned at 21 days of age, and approximately 7 weeks later, they were subjected to orthotopic tumor implantation. A total of 120 mice were used in this study. All mice were housed in a barrier facility under specific pathogen-free conditions at a controlled temperature of 23±2°C with 55±10% humidity and 12 h light/12 h dark light cycles. The animals were maintained on a sterilized standard diet and water. Panc02 variant cells ($2 \times 10^5$ cells) were orthotopically implanted into mice (4~7 mice per group). For this, the cells were implanted with 50% low-temperature Matrigel into the parenchyma of pancreatic tails of anesthetized mice with medetomidine (0.3 mg/kg)/midazolam (4.0 mg/kg)/butorphanol (5.0 mg/kg). The pancreas and spleen were exteriorized through a laparotomy, and the cells were injected using a 30-gauge needle attached to an insulin syringe. The mice and surgical wounds were observed and evaluated at least once a day for a minimum of one week until the mice returned to normal behavior and physical condition. Thereafter, the mice were examined daily for their health, including infection, wound dehiscence and excessive weight loss, and, if necessary, the mice received subcutaneous administration of analgesics to minimize pain and distress. At the end of a study, the mice were euthanized by continued $CO_2$ inhalation for at least 15 min after respiratory arrest following an IACUC approved protocol in which humane study endpoint criteria were described. In this study, the mice were euthanized before obvious distress, failure to ambulate, or weight loss > 10% was evident. Tumors were excised and weighed. Tumor volume (V) was calculated using the following equation: $V = (a^2 \times b)/2$, where a is the small diameter and b the large diameter.

## SB225002 administration

To investigate the effect of SB225002 on tumor growth, C57BL/6J mice were orthotopically implanted with Panc02-shCont cells ($2\times10^5$ cells in 50% Matrigel). Three days after injection, the mice were randomly (simple randomization) and blindly grouped into the vehicle (n = 11 mice) and SB225002 groups (n = 12 mice). In the SB225002 group, the mice were intraperitoneally administered 2 mg/kg SB225002 dissolved in 1% DMSO/10% PEG300/0.33% Tween 80/DPBS with single daily dosing. In the vehicle group, the mice were administered vehicle alone. Tumors were resected and weighed on Day 21 after Panc02 implantation.

## Immunohistochemical analysis of CD31 using frozen sections

Surgically removed Panc02 tumors were embedded and frozen in OCT compound. Cryostat sections (8-μm thick) were fixed in 4% paraformaldehyde for 10 min and blocked with 1% BSA in DPBS. The sections were incubated with rat anti-mouse monoclonal CD31 antibody (1:100 dilution) (BD Biosciences, Franklin Lake, NJ, USA) followed by incubation with Alexa Fluor 594-conjugated goat anti-rat IgG (1:300 dilution) (Thermo Fisher Scientific) and counterstaining with DAPI. Images were taken with a confocal laser scanning microscope (TCS SC8, Leica Microsystems, Wetzlar, Germany), and pixel values of the CD31-positive areas were calculated for each image to determine the tumor vessel density using ImageJ software (National Institutes of Health).

## Immunohistochemical analysis of paraffin-embedded tissues

Tumors were surgically removed and fixed in 10% phosphate-buffered formalin solution. The tissues were embedded in paraffin and cut into 2-μm sections. After heat-induced antigen retrieval in REAL Target Retrieval Solution (DAKO) and subsequent blocking of nonspecific sites with 0.1% normal goat serum/1% BSA, the tissues were immunostained with rabbit polyclonal anti-myeloperoxidase (MPO) antibody (Bioss Inc., Boston, MA, USA) diluted 1:200 using a previously described method [22]. The tissues were also immunostained with rabbit polyclonal anti-MPO antibody or rabbit polyclonal anti-CD20 antibody (Thermo Fisher Scientific) diluted 1:200, followed by Alexa Fluor 594-conjugated goat anti-rat IgG or phycoerythrin (PE)-conjugated rat monoclonal anti-mouse F4/80 (Abcam) diluted 1:100. In some experiments, after the tissues were treated with M.O.M. (Mouse on Mouse) Immunodetection Kit-Fluorescein (Vector Laboratories, Inc., Burlingame, CA, USA), the tissues were double immunostained with 10 μg/ml goat anti-mouse IL-33 IgG (R&D Systems, Minneapolis, MN, USA) and mouse monoclonal anti-α-SMA antibody diluted 1:100 (DAKO), followed by Alexa Fluor 594-conjugated chicken anti-goat IgG and Alexa Fluor 488-conjugated goat anti-mouse IgG. The sections were treated with the Vector TrueVIEW Autofluorescence Quenching Kit (Vector Laboratories) and counterstained with DAPI.

## Quantification of CXCL3 in tumor tissues by ELISA

Tumor tissue lysates were prepared as previously described [22]. Mouse CXCL3 concentrations were measured with the Cxcl3 ELISA Kit (Mouse) (Aviva Systems Biology, Corp. San Diego, CA, USA) according to the manufacturer's instructions.

## Statistical analysis

All data are presented as the mean ± SD. Statistical significance was tested using one-way ANOVA accompanied by pairwise comparisons with a *t*-test. Statistical significance in tumor

weights between two groups was tested by the Mann-Whitney U test. A value of $P<0.05$ was considered significant.

## Results

### Association between *IL33* and *ST2* expression and prognosis in pancreatic cancer patient databases

To examine the association between *IL33* and *ST2* (*IL1RL1*) expression and prognosis in pancreatic cancer patients, we searched the public prognosis databases PROGgeneV2 (http://watson.compbio.iupui.edu/chirayu/proggene/database/?url=proggene) and SurvExpress (http://bioinformatica.mty.itesm.mx:8080/Biomatec/SurvivaX.jsp). The datasets showed a trend toward poorer overall survival (OS) in patients with low expression of *IL33* than in patients with high expression in two out of three datasets (S1 Fig). On the other hand, the association between *ST2* expression and overall survival was inconsistent; patients with low *ST2* expression showed poorer OS than those with high expression in one dataset (GSE57495), whereas in other datasets, the association was the complete opposite (GSE71729) or less relevant (PAAD-TCGA) (S1 Fig). Thus, we could not find a consistent association between *IL-33* and *ST2* expression and prognosis. However, because IL-33-ST2L signaling is modulated by the amount of sST2 and microarray analyses measure only *ST2L* or do not draw a distinction between *ST2L* and *sST2*, we thought that evaluating the effect of sST2 released from cancer cells during tumor growth is important for proper prognostic analyses. This prompted us to embark on a study to reveal the role of sST2 in the tumor microenvironment of pancreatic cancer.

First, we measured the expression of *sST2* in six human pancreatic cancer cell lines and found that two cell lines (BxPC3 and CFPAC-1) expressed a large amount of *sST2* mRNA, while the other cell lines expressed little sST2 mRNA. The expression of *ST2L* mRNA was quite low in all cell lines (Fig 1A, S2 Fig). These results suggest that the amount of sST2 in pancreatic cancer tissues may be considerably different in every patient, which might bias the analyses of the association between IL-33 and ST2L expression and prognosis.

### Downregulation of sST2 expression in Panc02 cells inhibits orthotopic tumor growth

We assumed that the change in the expression level of sST2 in pancreatic cancer cells might affect tumor growth by influencing IL-33-ST2L signaling in the tumor microenvironment. To test this possibility, we employed an orthotopic implantation model in immunocompetent mice using Panc02 cells that express sST2 (Fig 1B). Panc02 cells expressed very little *IL-33* mRNA, and Panc02 variant cells (see below) did not secrete IL-33 protein (S3 Fig). Because IL-33 is reported to induce apoptosis in human pancreatic cancer MIAPaCa-2 cells [26], we first checked the effect of IL-33 on the growth of Panc02 cells and confirmed that IL-33 did not induce cell death in these cells (S4 Fig). We then knocked down ST2 expression by shRNA (sh#3 for knockdown of both ST2L and sST2 and sh#5 for knockdown of sST2 only). As a result, we established Panc02-sh#3 cells and Panc02-sh#5 cells that showed a reduced level of sST2 compared to that of Panc02-shCont cells at both the mRNA and the protein levels (Fig 2A and 2B). The *in vitro* growth of both Panc02-sh#3 and Panc02-sh#5 cells was comparable to that of Panc02-shCont cells (Fig 2C). However, when these cells were injected into the pancreas of C57BL/6J mice, both Panc02-sh#3 and Panc02-sh#5 cells formed significantly smaller tumors than Panc02-shCont cells (Fig 2D–2F).

Because sST2 was commonly suppressed in Panc02-sh#3 and Panc02-sh#5 cells, the suppression of tumor growth seemed to be due to sST2 depletion. To confirm this, we first

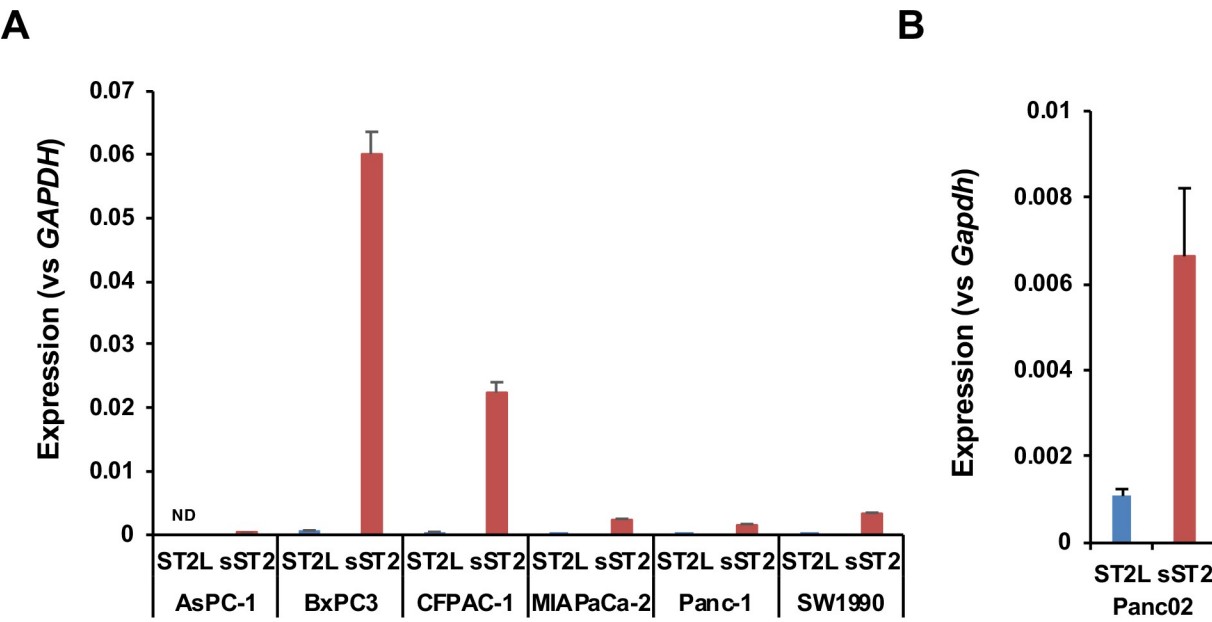

**Fig 1. Expression of ST2L and sST2 in pancreatic cancer cell lines.** qRT-PCR analysis of the expression of *ST2L* and *sST2*. The expression level was normalized to that of *GAPDH*. (A) Human pancreatic cancer cell lines. (B) Mouse pancreatic cancer Panc02 cells.

measured sST2 and IL-33 in the sera of tumor-bearing mice. The results showed that the amount of serum sST2 was much less in the mice bearing Panc02-sh#3 and Panc02-sh#5 tumors than in the mice bearing Panc02-shCont tumors, whereas the IL-33 level was comparable among them (S5 Fig). Next, we transfected the expression vector harboring *sST2* cDNA that lacks the #5 ST2 shRNA target sequence (pcDNA3.1/sST2Δ3'UTR) into Panc02-sh#5 cells. The resulting cells, Panc02-sh#5-sST2, expressed more sST2 at both the mRNA and the protein levels than Panc02-sh#5-VC cells that had been transfected with the vector alone (Fig 3A and 3B). Panc02-sh#5-sST2 cells grew slightly slower than Panc02-sh#5-VC and Panc02-shCont cells *in vitro* (Fig 3C). As expected, when the cells were injected into the pancreas of C57BL/6J mice, Panc02-sh#5-sST2 cells developed larger tumors than Panc02-sh#5-VC cells (Fig 3D–3F). However, Panc02-sh#5-sST2 tumors were smaller than Panc02-shCont tumors; the reason for this was unknown but could be due to insufficient recovery of sST2 or undesired effects of plasmid introduction and G418 selection. We also injected Panc02-shCont and Panc02-sh#5 cells into the pancreas of C57BL/6N and C57BL/6N-IL-33$^{-/-}$ (IL-33 KO) mice. Again, Panc02-sh#5 cells formed smaller tumors than Panc02-shCont cells in C57BL/6N mice. However, they formed tumors of equivalent size in IL-33 KO mice (Fig 3G–3I), indicating the involvement of IL-33 signaling in the difference in orthotopic tumor growth between Panc02-shCont and Panc02-sh#5 cells in wild-type mice. Thus, these results indicate that sST2 enhances orthotopic tumor growth of Panc02 cells by inhibiting the IL-33 response in the tumor microenvironment.

## Gene expression of cytokines and chemokines in orthotopic tumors

We compared the levels of *IL-33* and *ST2L* mRNA in Panc02-sh#5 tumors to those in Panc02-shCont tumors and found that they were nearly equal (Fig 4A). Immunohistochemical staining revealed that IL-33 protein as mostly localized in the cytoplasm of α-SMA-positive cells, which were probably pancreatic stellate cells or cancer-associated fibroblasts in the tumor microenvironment (S6 Fig), suggesting that these cells are the major source of secretory

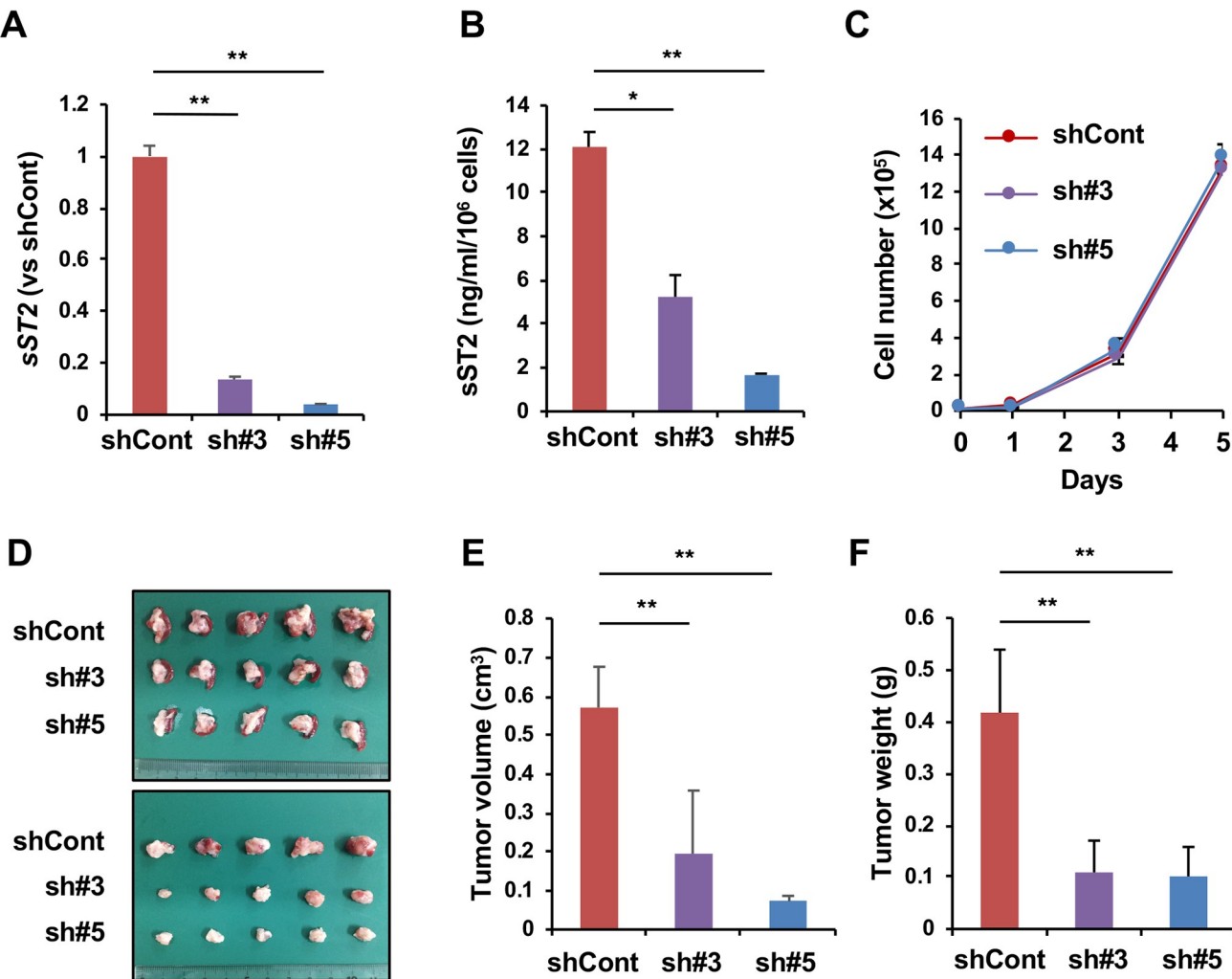

**Fig 2. sST2 downregulation suppresses orthotopic tumor growth of Panc02 cells.** (A) Knockdown of sST2 expression by ST2 shRNA. The expression levels of sST2 in Panc02-shCont, Panc02-sh#3 and Panc02-sh#5 cells were normalized to the expression of *Gapdh* and are presented as the ratio to the expression in Panc02-shCont cells (n = 3). (B) Secretion of sST2 by Panc02-shCont (n = 3), Panc02-sh#3 (n = 3) or Panc02-sh#5 cells (n = 3). (C) *In vitro* growth of Panc02-shCont, Panc02-sh#3 or Panc02-sh#5 cells (n = 3). (D) Orthotopic tumor growth of Panc02-shCont, Panc02-sh#3 and Panc02-sh#5 cells. The cells were implanted into the pancreas of C57BL/6J mice. Three weeks later, the pancreas and spleen (upper) were removed, and the tumors were excised (lower). (E) Orthotopic tumor volume formed by Panc02-shCont (n = 5 mice), Panc02-sh#3 (n = 6 mice) and Panc02-sh#5 cells (n = 7 mice). (F) Orthotopic tumor weight. Similar results were obtained in three independent experiments. The data are shown as the mean ± SD. *P<0.05. **P<0.001.

IL-33 protein. Then, to determine possible mechanisms underlying the difference in orthotopic tumor growth between Panc02-shCont and Panc02-sh#5 cells, we compared the cytokine and chemokine gene expression profiles between the Panc02-shCont and Panc02-sh#5 tumors using a PCR array platform. The results showed that the expression of *Cxcl3*, which encodes a neutrophil chemoattractant, was significantly downregulated in Panc02-sh#5 tumors (Fig 4B). By using RT-qPCR analysis, we found that among the members of the CXC chemokine family, *Cxcl2*, *Cxcl3* and *Cxcl5* showed relatively abundant expression in orthotopic tumors, but only *Cxcl3* was significantly downregulated in Panc02-sh#5 tumors (Fig 4C). There was no difference in the expression of *Cxcr2*, the receptor for these chemokines (Fig 4C). Because these chemokines had very little expression in Panc02 cells (Fig 4C), they seemed to derive from stromal cells in the tumor microenvironment. It may be worthwhile to note that Oncomine

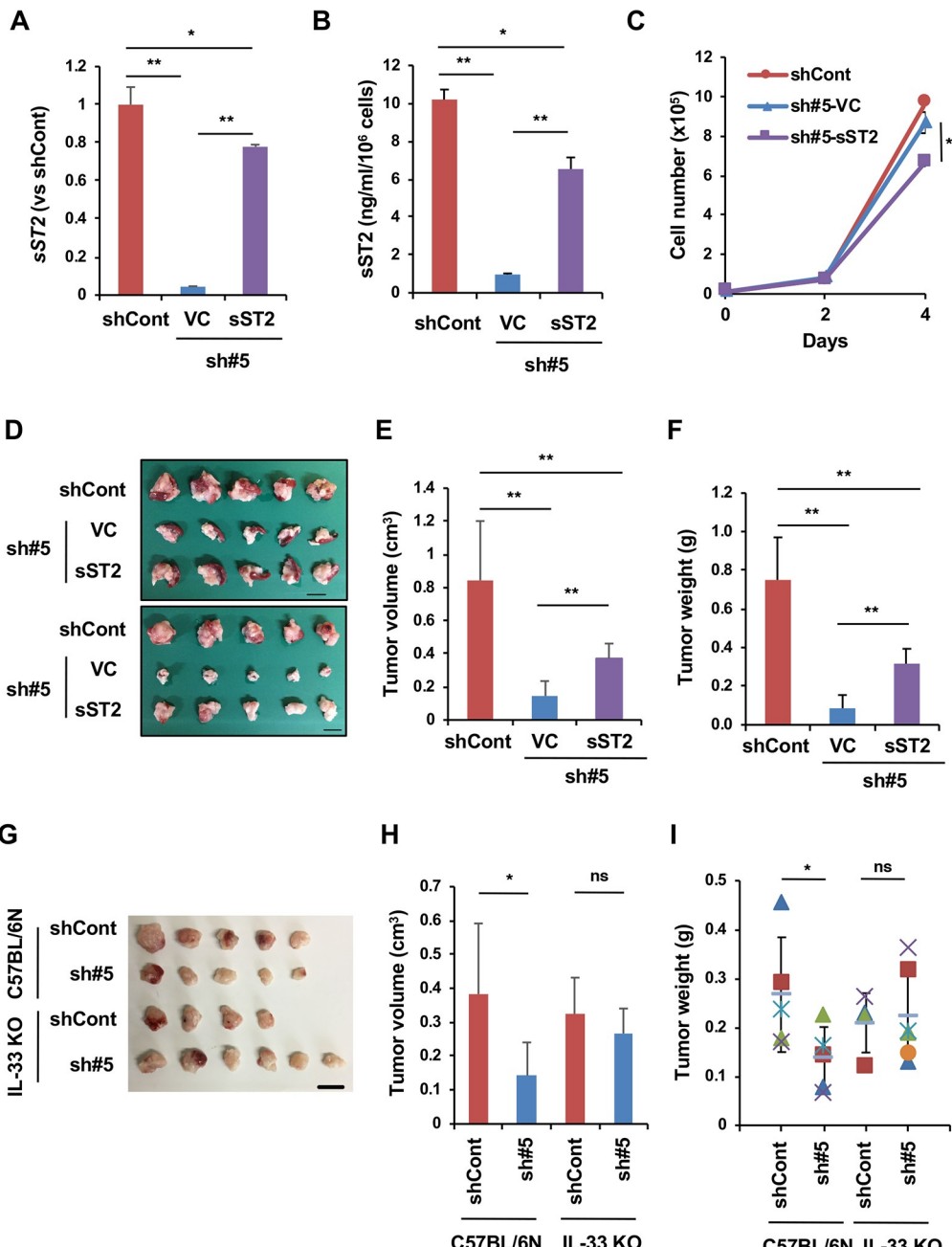

**Fig 3. Suppression of orthotopic tumor growth of Panc02-sh#5 cells is dependent on sST2 and IL-33.** (A) Establishment of Panc02-sh#5 cells that re-express sST2. RT-qPCR analysis of *sST2* in Panc02-shCont (n = 3), Panc02-sh#5-VC (n = 3) and Panc02-sh#5-sST2 cells (n = 3). The expression levels are normalized to the expression of *Gapdh* and are presented as the ratio to the expression in Panc02-shCont cells. (B) Secretion of sST2 by Panc02-shCont (n = 3), Panc02-sh#5-VC (n = 3) or Panc02-sh#5-sST2 cells (n = 3). (C) *In vitro* growth of Panc02-shCont (n = 3), Panc02-sh#5-VC (n = 3) or Panc02-sh#5-sST2 cells (n = 3). (D) Orthotopic tumor growth of Panc02-shCont, Panc02-sh#5-VC and Panc02-sh#5-sST2 cells. The cells were implanted into the pancreas of C57BL/6J mice. Three weeks later, the pancreas and spleen (upper) were removed, and the tumors were excised (lower). Bar: 1 cm. (E) Orthotopic tumor volume formed by Panc02-shCont (n = 5 mice), Panc02-sh#5-VC (n = 7 mice) and Panc02-sh#5-sST2 cells (n = 6 mice). (F) Orthotopic tumor weight. (G) Orthotopic tumor growth of Panc02-shCont and Panc02-sh#5 cells in C57BL/6N and C57BL/6N-IL-33$^{-/-}$ mice. The cells were implanted into the pancreas of C57BL/6N (n = 5 mice for Panc02-shCont cells, n = 5 mice for Panc02-sh#5 cells) or C57BL/6N-IL-33-/- mice (n = 4 mice for Panc02-shCont cells, n = 6 mice for Panc02-sh#5 cells). Bar: 1 cm. (H) Orthotopic tumor volume formed by Panc02-shCont cells and Panc02-sh#5 cells in C57BL/6N (n = 5 mice for Panc02-shCont cells, n = 5 mice for

Panc02-sh#5 cells) or C57BL/6N-IL-33-/- (IL-33 KO) mice (n = 4 mice for Panc02-shCont cells, n = 6 mice for Panc02-sh#5 cells). (I) Orthotopic tumor weight. *P<0.05, **P<0.001. ns, not significant.

datasets showed that *Cxcl3* expression was relatively high in human pancreatic adenocarcinomas (https://www.oncomine.org/resource/login.html) (S7 Fig). On the other hand, *Cxcl13*, which encodes a B cell chemoattractant [30], was the most upregulated gene in Panc02-sh#5 tumors in the PCR array analysis (Fig 4B). However, this was not statistically significant in the RT-qPCR analysis because the expression level varied considerably among tumor samples (Fig 4D).

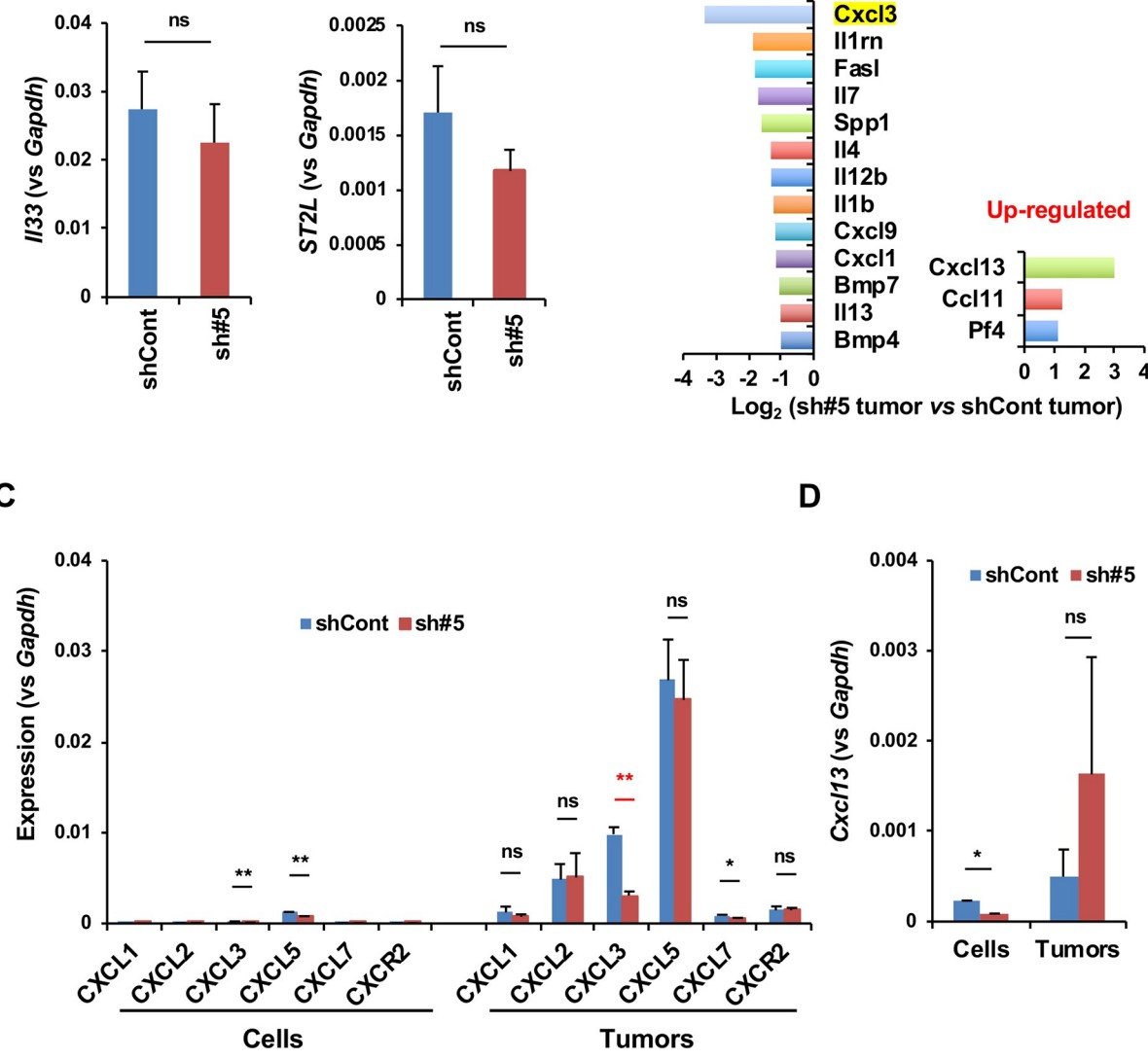

**Fig 4. Expression of IL-33, ST2L, cytokines and chemokines in orthotopic tumors.** (A) RT-qPCR analysis of *Il33* and *ST2L* in Panc02-shCont and Panc02-sh#5 tumors (n = 3). (B) Gene expression profiling of cytokines and chemokines between Panc02-shCont and Panc02-sh#5 tumors were performed using a PCR array. (C) RT-qPCR analysis of various chemokine (C-X-C motif) ligands and *Cxcr2* in Panc02-shCont and Panc02-sh#5 cells (n = 3) and tumors (n = 7–9). (D) RT-qPCR analysis of *Cxcl13* in Panc02-shCont (n = 7) and Panc02-sh#5 tumors (n = 9). The data are shown as the mean ± SD. *P<0.05. **P<0.001. ns, not significant.

To examine whether IL-33 influences the expression of sST2 and ST2L in Panc02 cells, we treated the cells with various concentrations of rIL-33 for up to 24 h. As a result, we found that rIL-33 slightly but significantly increased the mRNA expression of both sST2 and ST2L (S8 Fig). These results suggest that IL-33 in the tumor microenvironment enhances the secretion of sST2 from tumor cells, which in turn enhances orthotopic tumor growth. We further examined the effect of rIL-33 on the expression of *Cxcl3* in Panc02 cells. However, the basal expression level of *Cxcl3* in the cells was very low, although rIL-33 only slightly increased *Cxcl3* expression (S9 Fig).

## CXCL3-mediated infiltration of neutrophils and microvessel density in orthotopic tumors

ELISA analysis also confirmed the lower expression of CXCL3 in Panc02-sh#5 tumors than in Panc02-shCont tumors (Fig 5A). CXCL3 is known to recruit and activate neutrophils and enhance angiogenesis by binding to its receptor, CXCR2, which is expressed mainly on neutrophils and endothelial cells [31–33]. Then, we compared neutrophil infiltration and microvessel density between Panc02-shCont and Panc02-sh#5 orthotopic tumors. Immunostaining of the neutrophil marker MPO showed few MPO$^+$ cells in Panc02-sh#5 tumors compared to those in Panc02-shCont tumors (Fig 5B and 5C). RT-qPCR analysis showed that the expression of another neutrophil marker gene, *Ly6G*, was also lower in Panc02-sh#5 cells than in Panc02-shCont cells (Fig 5D). However, the tumor infiltration of F4/80$^+$ macrophages and CD20$^+$ B cells was not significantly different between the Panc02-shCont and Panc02-sh#5 tumors (Fig 5B, 5E and 5F). CD31 immunostaining revealed a significantly lower vessel density in Panc02-sh#5 tumors than in Panc02-shCont tumors (Fig 5G and 5H).

## Inhibition of CXCR2 signaling suppresses orthotopic tumor growth

To confirm whether CXCR2 signaling is involved in Panc02 orthotopic tumor growth, we examined the effect of the CXCR2 antagonist SB225002. For this purpose, SB225002 was administered intraperitoneally every day at a dose of 2 mg/kg for 16 days to mice bearing Panc02-shCont orthotopic tumors (Fig 6A). The results showed that SB225002 suppressed orthotopic tumor growth without body weight loss (Fig 6B–6E). Furthermore, both neutrophil infiltration (Fig 6F and 6G) and vessel density (Fig 6H and 6I) in SB225002-treated tumors were significantly lower than those in control tumors. Thus, these results suggest that the CXCL3/CXCR2 axis is at least responsible for neutrophil infiltration, tumor angiogenesis and slowed orthotopic tumor growth and may be involved in the difference in orthotopic tumor growth between Panc02-shCont and Panc02-sh#5 cells.

## Discussion

We described that sST2 downregulation in Panc02 cells resulted in suppression of tumor growth in the pancreas. This effect was dependent on host-derived IL-33 because Panc02 cells do not secrete IL-33, and we did not detect the suppressive effect in IL-33 knockout mice. Although IL-33 has been shown to induce apoptosis in MIAPaCa-2 cells [26], we did not detect cell death in rIL-33-treated Panc02-shCont cells. Therefore, it seemed likely that the decrease in sST2 in the tumor microenvironment led to activation of IL-33/ST2L signaling, leading to either activation or suppression of other genes in stromal cells. One such gene seemed to be *Cxcl3*, which was identified by analysis of the expression of cytokines and chemokines; that is, *Cxcl3* expression was suppressed in sST2-downregulated orthotopic tumor tissues. CXCL3 has been shown to bind to the receptor CXCR2 and thereby activate migration of neutrophils and endothelial cells [34–38]. Accordingly, immunostaining of the tumor tissues

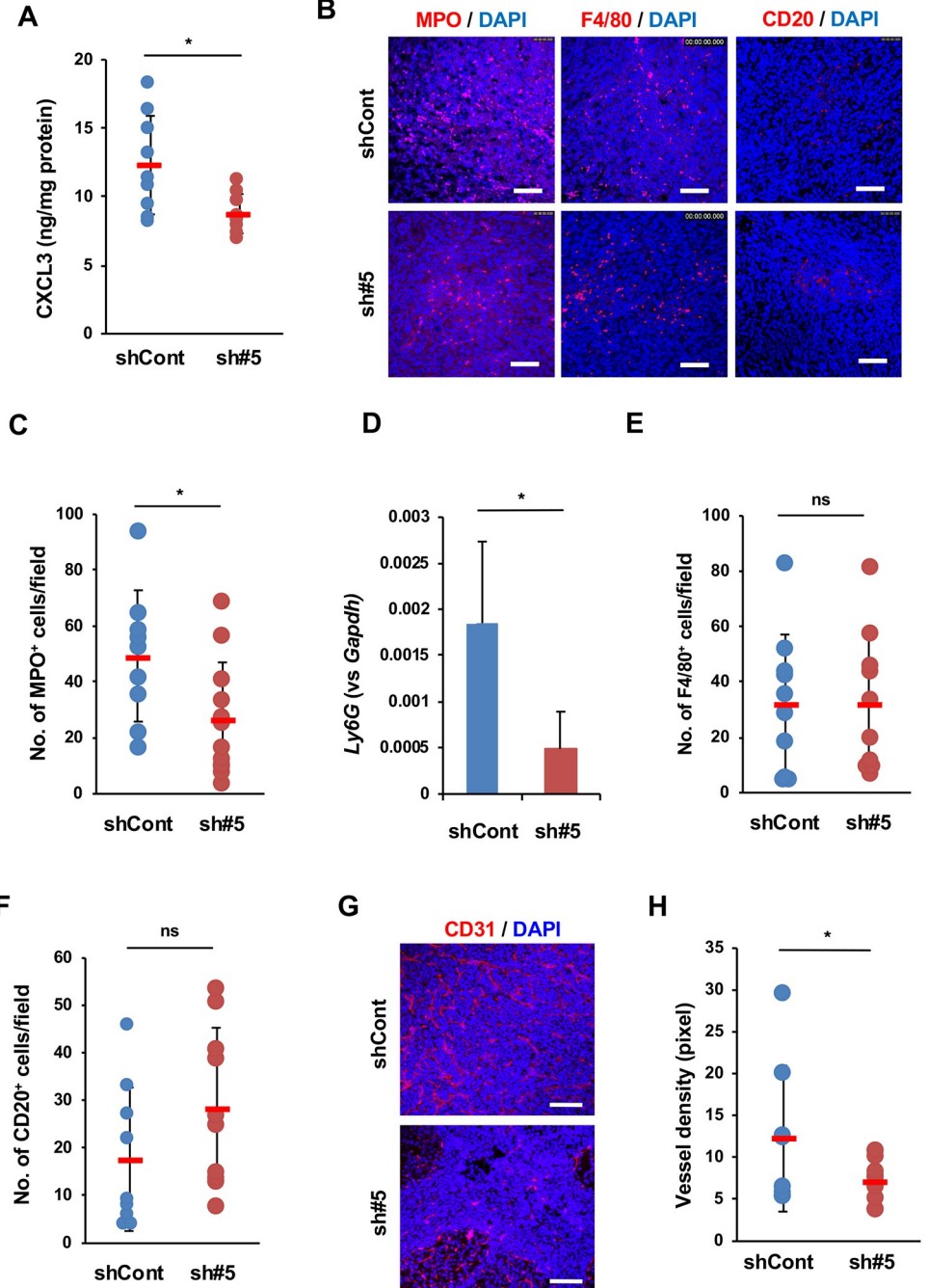

**Fig 5. Infiltration of neutrophils and tumor angiogenesis in orthotopic tumors.** (A) The amount of CXCL3 in Panc02-shCont (n = 9) and Panc02-sh#5 (n = 9) tumor tissues. (B) IHC analysis of MPO[+] neutrophil infiltration in Panc02-shCont and Panc02-sh#5 tumors. Bar: 50 μm. (C) The number of MPO[+] cells per field (×400 magnification) in Panc02-shCont (n = 9 fields of 2 tissue sections) and Panc02-sh#5 tumors (n = 12 fields of 2 tissue sections). (D) RT-qPCR analysis of expression of the neutrophil marker gene *Ly6G* in Panc02-shCont (n = 5 tumors) and Panc02-sh#5 tumors (n = 6 tumors). (E) The number of F4/80[+] cells per field (×400 magnification) in Panc02-shCont (n = 10 fields of 5 tissue sections) and Panc02-sh#5 tumors (n = 12 fields of 4 tissue sections). (F) The number of CD20[+] cells per field (×400 magnification) in Panc02-shCont (n = 9 fields of 3 tissue sections) and Panc02-sh#5 tumors (n = 10 fields of 3 tissue sections). (G, H) Angiogenesis in Panc02-shCont and Panc02-sh#5 tumors. G: CD31 staining. Bar: 50 μm. H: Vessel density. Panc02-shCont (n = 11 fields) and Panc02-sh#5 cells (n = 14 fields). The data are shown as the mean ± SD. *P<0.05. ns, not significant.

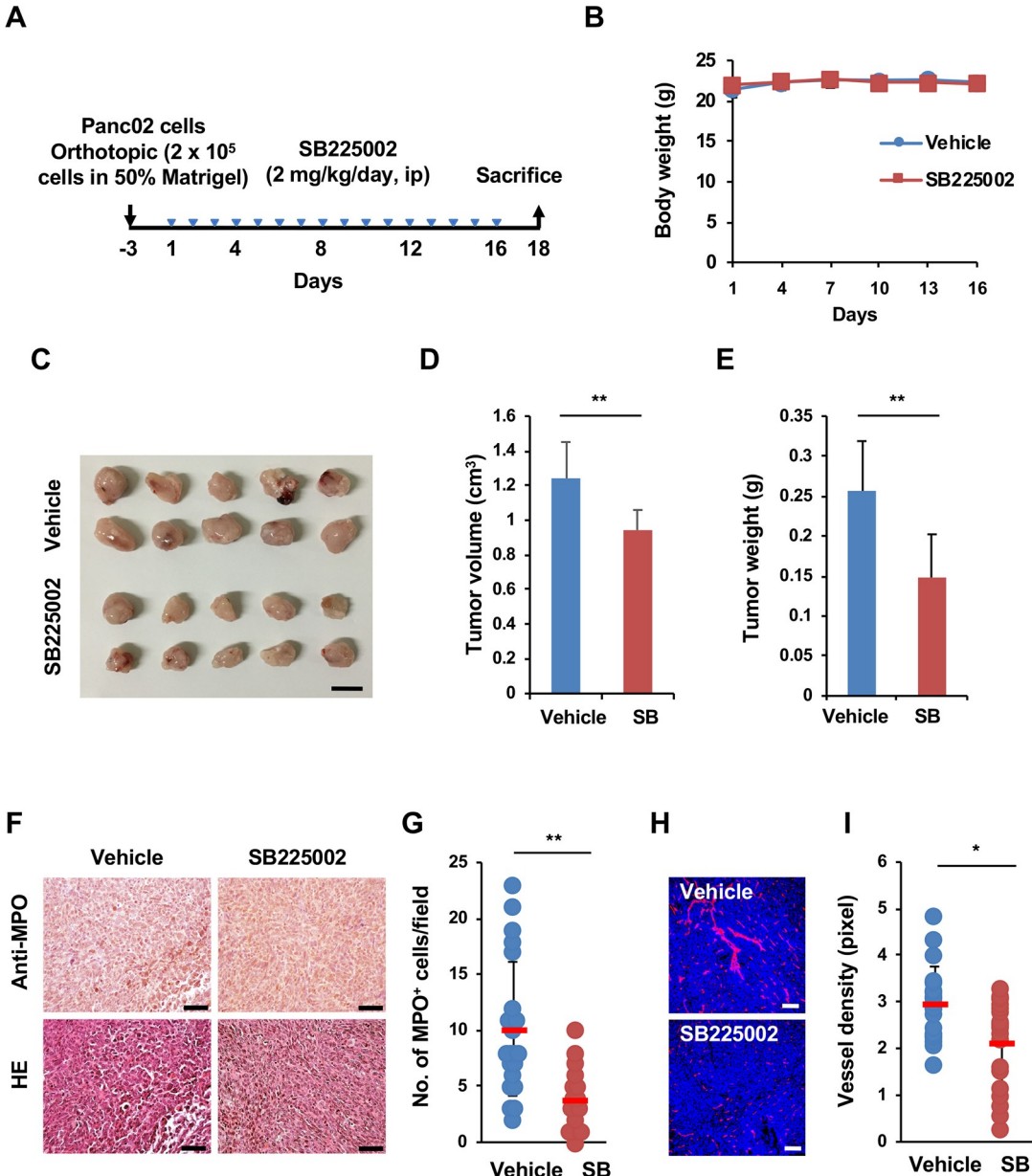

**Fig 6. Effect of SB225002 administration on tumor growth, neutrophil infiltration and angiogenesis.** (A) Protocol used for SB225002 administration. Panc02 cells ($2 \times 10^5$ cells in 50% Matrigel) were inoculated into the pancreas. Three days after inoculation, SB225002 (2 mg/kg) (n = 12 mice per group) or vehicle alone (n = 11 mice per group) was administered intraperitoneally every day for 16 days. The mice were euthanized, and the tumors were removed on Day 18. (B) Body weight. (C) Orthotopic tumors. Bar: 1 cm. (D) Tumor volume. (E) Tumor weight. (F) IHC analysis of MPO$^+$ neutrophil infiltration in Panc02-shCont and SB225002-treated Panc02-shCont tumors. Bar: 50 μm. (G) The number of MPO$^+$ cells per field (×400 magnification) in Panc02-shCont (n = 22 fields of 4 tissue sections) and Panc02-sh#5 tumors (n = 17 fields of 5 tissue sections). (H) Angiogenesis. CD31 staining. Bar: 100 μm. (I) Tumor vessel density in Panc02-shCont and SB225002-treated Panc02-shCont tumors (n = 20 fields of 6–8 tissue sections). The data are shown as the mean ± SD. $^*P<0.05$. $^{**}P<0.01$.

for MPO showed a decrease in the number of MPO$^+$ cells in Panc02-sh#5 tumors compared with that of Panc02-shCont tumors. The decrease in neutrophil infiltration was also supported by reduced expression of the neutrophil marker gene *Ly6G* in Panc02-sh#5 tumors. Although the role of neutrophils in cancer is still controversial, the presence of high numbers of tumor-

associated neutrophils (TANs) has been reported to be associated with worse outcomes in several solid tumors [38–42]. Therefore, the decrease in neutrophils in Panc02-sh#5 tumors may be associated with the slow orthotopic growth of the tumors. Furthermore, CD31 staining revealed a decrease in tumor vessel density, which may also be responsible for the slow orthotopic growth. These results were corroborated by SB225002 administration experiments in which SB225002 slowed orthotopic tumor growth of Panc02-shCont cells, accompanied by decreases in neutrophil infiltration and vessel density. It should be noted that the number of TANs in Panc02-sh#5 tumors was not different from that in Panc02-shCont tumors, which quite contrasted with the results obtained in colon cancers [22].

It remains to be determined how sST2 downregulation leads to a decrease in *Cxcl3* expression. A previous study showed that IL-33 slightly downregulates *Cxcl3* expression in human alveolar type-II-like A549 cells [43]. We observed here that rIL-33 only slightly enhanced *Cxcl3* expression in Panc02 cells. Based on these results, it is plausible that IL-33 secreted from stellate cells or CAFs acts on other cells, such as macrophages, that are reported to express ST2L and CXCL3 [7–9, 44], rather than on tumor cells themselves to stimulate the secretion of CXCL3. It is possible that a decrease in sST2 in the orthotopic tumor microenvironment activates IL-33/ST2L signaling in stellate cell-TAM or CAF-TAM interactions, which results in the suppression of *Cxcl3* expression. Further studies are necessary to evaluate this possibility.

We found that CXCL5 was abundantly expressed in orthotopic tumors. CXCL1, CXCL2, CXCL3 and CXCL5 all bind to CXCR2 [45]. Thus, one can argue that CXCL3 downregulation is of no importance in the presence of a large amount of CXCL5. With regard to this point, it is noteworthy that CXCL5 is reported to bind to the Duffy antigen receptor for chemokines (DARC) that is expressed in endothelial cells and erythrocytes and acts as a decoy chemokine receptor [45, 46]. DARC has a higher affinity for CXCL5 than other neutrophil chemokines, such as CXCL1 and CXCL2, and therefore, when all of these chemokines are present, CXCL5 remains bound to DARC, leaving CXCL1 and CXCL2 free and active [46]. If such is the case with CXCL3, it could act as the main chemokine to recruit neutrophils and stimulate angiogenesis in the tumors because CXCL3 is more abundant than CXCL1 and CXCL2. Further studies are needed to examine this issue.

On the other hand, a marked increase in the expression of the B cell chemoattractant *Cxcl13* in Panc02-sh#5 tumors was evident when examined by a PCR array. However, RT-qPCR analysis showed that the expression varied significantly from tumor tissue to tumor tissue; therefore, no statistically significant difference was obtained between Panc02-shCont and Panc02-sh#5 tumors. Accordingly, we did not observe a significant difference in the number of tumor-infiltrating B cells between the Panc02-sh#5 and Panc02-shCont tumors. We hypothesized that this variation might be due to the region of the tumor tissues from which the RNA was isolated, in which B cells could show skewed distribution.

We previously reported that sST2-knockdown mouse colon cancer cells are less malignant than control cells [22, 47]. Therefore, the present findings are unexpected. However, recent reports have demonstrated that the pancreatic but not subcutaneous tumor microenvironment preferentially confers highly malignant properties to pancreatic cancer cells [48], and gene expression profiles in pancreatic cancer are different between orthotopic and subcutaneous conditions [49]. These data indicate that the microenvironment in the pancreas is composed of a variety of components, including pancreatic stellate cells, endothelial cells, immune cells and endocrine cells, and is very different from that in other organs, and it may profoundly affect the malignant properties of sST2-expressing pancreatic cancer cells. However, the limitation of this study is that we utilized an orthotopic implantation model, which could be a potential source of bias in interpreting the results. In addition, because there are few murine pancreatic cancer cell lines that are well characterized and can be investigated from an

immunological point of view in syngeneic mice, the present findings remain generalizable to other pancreatic cancer cell lines in general. Further studies using a spontaneous pancreatic cancer mouse model in which the *Il33* or *ST2* (*IL1RL1*) gene is genetically engineered are required in the future.

## Conclusions

In the public prognosis databases, we could not find a consistent association between ST2 expression and the prognosis of pancreatic cancer patients. We hypothesized that this may be attributed to a lack of consideration of the effect of sST2 on pancreatic cancer growth. In fact, the present study demonstrated that activation of the IL-33/ST2L axis by suppressing sST2 in the tumor microenvironment inhibited orthotopic tumor growth of Panc02 cells. These results suggest that targeting sST2 in the tumor microenvironment might inhibit tumor growth of sST2-expressing human pancreatic cancers, which is worth exploring in the future. Our results also indicate the CXCL3-CXCR2 axis as a therapeutic target.

## Supporting information

**S1 Table. Primers used for qRT-PCR.**
(TIF)

**S1 Fig. Correlation of *IL-33* and *ST2* (*IL1RL1*) gene expression in cancer tissues and overall survival of pancreatic cancer patients.**
(TIF)

**S2 Fig. qRT-PCR analysis of the expression of *ST2L* and *sST2*.** The expression level is expressed as ΔCt. (A) Human pancreatic cancer cell lines. (B) Mouse pancreatic cancer Panc02 cells. moST2L: mouse ST2L; mosST2: mouse sST2. ND: not detected.
(TIF)

**S3 Fig. Expression and secretion of IL-33 in Panc02 parent and variant cells.** Expression and secretion of IL-33 in Panc02 parent and variant cells. (A) qRT-PCR analysis of the expression of IL-33 mRNA in Panc02 cells and mouse colon carcinoma NM11 cells (Nat Commun. 2016;7:13589) (left panel) in which IL-33 is localized in the nucleus (right panel). (B) Secretion of IL-33. Cells ($4 \times 10^4$ cells/100 μl) were cultured for 24 h, and the conditioned media were subjected to IL-33 ELISA analysis. The amount of IL-33 was below the detection limits.
(TIF)

**S4 Fig. Effect of rIL-33 on the survival of Panc02 cells.** The cells were treated with the indicated concentrations of rIL-33 for 48 h. An MTT assay was used to measure cell survival.
(TIF)

**S5 Fig. Serum sST2 and IL-33 levels in mice bearing Panc02-shCont, Panc02-sh#3 and Panc02-sh#5 tumors.** (A) sST2. (B) IL-33. Sera of the mice bearing shCont tumors (n = 5), sh#3 tumors (n = 4) and sh#5 tumors (n = 4) were used. *P<0.001 compared to shCont.
(TIF)

**S6 Fig. IL-33 immunohistochemistry.** Sections of Panc02-shCont orthotopic tumors were immunostained for IL-33 and α-SMA. Bar: 50 μm.
(TIF)

**S7 Fig. Ramaswamy dataset of Cxcl3 expression in various human tumors (Oncomine).**
(TIF)

**S8 Fig. qRT-PCR analyses of the effect of rIL-33 on the expression of *sST2* and ST2L in Panc02 cells.** (A) The cells were treated with the indicated concentrations of rIL-33 for 6 h. (B) The cells were treated with 50 ng/ml rIL-33 for the indicated times. *P<0.05 compared to untreated cells.
(TIF)

**S9 Fig. qRT-PCR analyses of the effect of rIL-33 on the expression of *Cxcl3* in Panc02 cells.** (A) The cells were treated with the indicated concentrations of rIL-33 for 6 h. (B) The cells were treated with 50 ng/ml rIL-33 for the indicated times.
(TIF)

## Acknowledgments

Some animal experiments were performed in the animal center of Shimane University Faculty of Medicine.

## Author Contributions

**Conceptualization:** Keizo Takenaga.

**Data curation:** Keizo Takenaga, Miho Akimoto.

**Formal analysis:** Keizo Takenaga, Miho Akimoto.

**Funding acquisition:** Keizo Takenaga, Miho Akimoto.

**Investigation:** Keizo Takenaga, Miho Akimoto, Nobuko Koshikawa, Hiroki Nagase.

**Project administration:** Hiroki Nagase.

**Supervision:** Hiroki Nagase.

**Writing – original draft:** Keizo Takenaga.

**Writing – review & editing:** Keizo Takenaga.

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
