## [Decision Letter · Decision Letter 0]

27 Feb 2020

PONE-D-19-33979

Cancer cell-derived interleukin-33 decoy receptor sST2 enhances orthotopic tumor growth in a murine pancreatic cancer model

PLOS ONE

Dear Takenaga,

Thank you for submitting your manuscript to PLOS ONE. After careful consideration, we feel that it has merit but does not fully meet PLOS ONE’s publication criteria as it currently stands. Therefore, we invite you to submit a revised version of the manuscript that addresses the points raised during the review process.

We would appreciate receiving your revised manuscript by Apr 12 2020 11:59PM. To enhance the reproducibility of your results, we recommend that if applicable you deposit your laboratory protocols in protocols.io, where a protocol can be assigned its own identifier (DOI) such that it can be cited independently in the future. For instructions see: http://journals.plos.org/plosone/s/submission-guidelines#loc-laboratory-protocols

We look forward to receiving your revised manuscript.

Kind regards,

Sulagna Banerjee

Academic Editor

PLOS ONE

Journal Requirements:

3. Please provide additional information about each of the cell lines used in this work, including any quality control testing procedures (authentication, characterisation, and mycoplasma testing). For more information, please see http://journals.plos.org/plosone/s/submission-guidelines#loc-cell-lines.

Reviewers' comments:

Reviewer's Responses to Questions

**Comments to the Author**

1. Is the manuscript technically sound, and do the data support the conclusions?

Reviewer #1: Yes

Reviewer #2: Yes

2. Has the statistical analysis been performed appropriately and rigorously? 

Reviewer #1: Yes

Reviewer #2: Yes

3. Have the authors made all data underlying the findings in their manuscript fully available?

Reviewer #1: Yes

Reviewer #2: Yes

4. Is the manuscript presented in an intelligible fashion and written in standard English?

Reviewer #1: Yes

Reviewer #2: Yes

5. Review Comments to the Author

Reviewer #1: IL-33 has been widely studied in inflammatory diseases, but its role in cancer is somewhat paradoxical. Human pancreatic cancer is one of the deadliest malignancies wherein the inflammation of the pancreas plays a critical role. This research paper entitled “Cancer cell-derived interleukin-33 decoy receptor sST2 enhances orthotopic tumor growth in a murine pancreatic cancer model” encompasses some interesting findings wherein following issues should be addressed.

Minor Concerns:

1. Fig. 1 legend can be made more explanatory.

2. Fig. 4 legend need correction wherein last line saying *p<0.005 should be corrected as *p<0.05.

Major Concerns:

1. Fig.1: Authors should provide a plot for Δct values for ST2l and sST2 for all human and mouse cell lines in either Fig 1 or supplementary.

2. Fig 2: It would have been a better comparison to see the effect of knocking down the sST2 and ST2l individually instead of employing sh#3 shRNA which collectively knocks down both ST2l and sST2. Have authors used any shRNA to knock down ST2l individually? If so, please include the data to draw a better comparison and so the effect on tumor volume and tumor weight.

3. In Fig 3E and F the tumor volume and weight is approximately half fold in samples where sST2 is rescued as compared to control. Please discuss.

4. Fig 4A shows expression of IL-33 mRNA by RT-PCR. Since sST2 act as IL-33 sink, authors should show the expression of secretory IL-33 which is expected to up-regulate in sh#5 as compared to shCont.

Since idea of the paper is, Cancer cell-derived sST2 enhances tumor growth through upregulation of CXCL3 via inhibition of IL-33-ST2L signaling in the tumor microenvironment of pancreatic cancer, the expression of secretory IL-33 in sh#5 and shCont would be an essential piece of data to corroborate the finding in IL-33 knock out mice wherein decoy receptor seems to has no effect. Please include the said figure in Fig 4.

Reviewer #2: Here, Takenaga and colleagues conducted an in vivo study to explore the role of cancer cell derived IL-33 decoy receptor sST2 in pancreatic tumor growth. This is an interesting study as this one relatively new in pancreatic cancer and almost unexplored. The author precisely investigated the involvement of decoy receptor in tumor progression. However, in my opinion the manuscript has some shortcomings. Below I have provided numerous remarks. Please address the following concern.

Comments:

1. What is the source of IL-33 in TME. Is it pancreatic stellate cell or other cell types?

2. As a ligand, is IL-33 regulate the secretion of receptor sST2 from pancreatic cancer cell? If so, which control the secretion of receptor sST2 in global IL-33 KO mice. It is interesting to see that IL-33 signaling downregulate CXCL-3 expression, what could be the possible mechanism?

3. In figure3 panel E and F, is VC and sST2 are statistically significant?

4. It will be worthy, in future to perform some coculture experiment with PANC02 cell and CAF cell to better understand their cytokine/chemokine crosstalk in TME.

6. PLOS authors have the option to publish the peer review history of their article (what does this mean?). If published, this will include your full peer review and any attached files.

Reviewer #1: Yes: Utpreksha Vaish

Reviewer #2: Yes: kousik kumar kesh

---

## [Author Response · Author response to Decision Letter 0]

27 Mar 2020

Response to reviewers

Journal requirements

Reply: We have ensured that our manuscript meets the style requirements of PLOS ONE.

2. We suggest you thoroughly copyedit your manuscript for language usage, spelling, and grammar.

Reply: Our manuscript has been thoroughly edited by SPRINGER NATURE Author Services. We have attached the certificate.

3. Please provide additional information about each of the cell lines used in this work, including any quality control testing procedures (authentication, characterisation, and mycoplasma testing). 

Reply: We have provided additional information about Panc02 cells and mycoplasma testing (page 6, lines 106-108 and 113-115).

Reviewer #1

Major comments:

1. Fig. 1 legend can be made more explanatory.

Reply: Thank you for the comment. We have changed the legend.

2. Fig. 4 legend need correction wherein last line saying *p<0.005 should be corrected as *p<0.05.

Reply: We have corrected the p value.

Minor comments:

1. Fig.1: Authors should provide a plot for Δct values for ST2l and sST2 for all human and mouse cell lines in either Fig 1 or supplementary.

Reply: We have provided a plot showing the ΔCt values in the Supporting information (S1 Fig).

2. Fig 2: It would have been a better comparison to see the effect of knocking down the sST2 and ST2l individually instead of employing sh#3 shRNA which collectively knocks down both ST2l and sST2. Have authors used any shRNA to knock down ST2l individually? If so, please include the data to draw a better comparison and so the effect on tumor volume and tumor weight.

Reply: Unfortunately, shRNAs that specifically knock down only ST2L were not available at the time we started the experiments. Therefore, we did not compare the effect of individual knock down of ST2L and sST2.

3. In Fig 3E and F the tumor volume and weight is approximately half fold in samples where sST2 is rescued as compared to control. Please discuss.

Reply: As the reviewer pointed out, the growth of sh#5-sST2 tumors was not recovered to the level of shCont tumors. We do not know the reason, but it could be due to insufficient recovery of sST2 or off-target effects of the introduced expression by the plasmid and the G418 selection procedure of the introduced cells. We have discussed this point in the text (page 16, lines 323-325).

4. Fig 4A shows expression of IL-33 mRNA by RT-PCR. Since sST2 act as IL-33 sink, authors should show the expression of secretory IL-33 which is expected to up-regulate in sh#5 as compared to shCont. Since idea of the paper is, Cancer cell-derived sST2 enhances tumor growth through upregulation of CXCL3 via inhibition of IL-33-ST2L signaling in the tumor microenvironment of pancreatic cancer, the expression of secretory IL-33 in sh#5 and shCont would be an essential piece of data to corroborate the finding in IL-33 knock out mice wherein decoy receptor seems to has no effect. Please include the said figure in Fig 4.

Reply: We appreciate the comment very much. In separate experiments, we measured the levels of IL-33 and sST2 in the sera of mice bearing shCont, sh#3 and sh#5 tumors, as shown in Supporting information (S5 Fig). The results showed that serum sST2 levels were considerably decreased in the mice bearing sh#3 and sh#5 tumors compared to those bearing shCont tumors, whereas the levels of IL-33 were comparable among them. As the reviewer suggested, sST2 acts as an IL-33 sink, and thus, we expected that the IL-33 level would be increased in the sera of mice bearing sh#3 and sh#5 tumors. However, this was not the case. We hypothesize that this may be because the anti-IL-33 antibody used in the ELISA cannot differentiate free IL-33 from sST2-bound IL-33. Based on these observations, the comparison of IL-33 protein by ELISA between sh#5 tumors and shCont tumors does not provide concrete evidence indicating the role of sST2 as an IL-33 sink.

Reviewer #2

1. What is the source of IL-33 in TME. Is it pancreatic stellate cell or other cell types?

Reply: We appreciate the comment very much. We investigated the source of IL-33 in the TME by immunohistochemistry and found that α-smooth muscle actin+ cells were positive for IL-33. These cells could be pancreatic stellate cells or cancer-associated fibroblasts. The results are provided in the Supplementary information (S6 Fig).

2. As a ligand, is IL-33 regulate the secretion of receptor sST2 from pancreatic cancer cell? If so, which control the secretion of receptor sST2 in global IL-33 KO mice. It is interesting to see that IL-33 signaling downregulate CXCL-3 expression, what could be the possible mechanism?

Reply: We investigated the effects of rIL-33 on the expression of sST2 and ST2L in Panc02 cells and found that rIL-33 significantly increased the expression of both sST2 and ST2L (see Supplementary information S8 Fig). Because Panc02 cells do not secrete IL-33 (see Supplementary information S3 Fig), these results indicate that host-derived IL-33 partially regulates the expression of sST2 via IL-33-ST2L signaling. It has been shown that sST2 expression is regulated by oncogenes, serum and other mitogenic stimuli (Eur. J. Immunol. 2012. 42: 1863–1869). Therefore, in IL-33 KO mice, sST2 seems to be induced in response to these stimuli.

We also investigated the effect of rIL-33 on CXCL-3 expression in Panc02 cells. We found that Panc02 cells expressed little CXCL-3, and rIL-33 only slightly enhanced the expression (see Supplementary information S9 Fig). However, the increase seemed to be marginal. Therefore, IL-33 may act on stromal cells rather than tumor cells to enhance CXCL-3 secretion in the tumor microenvironment.

3. In figure 3 panel E and F, is VC and sST2 are statistically significant?

Reply: Thank you for the comment. Yes, the difference between VC and sST2 was statistically significant. We have included the p-value in Figure 3 panels E and F.

4. It will be worthy, in future to perform some coculture experiment with PANC02 cell and CAF cell to better understand their cytokine/chemokine crosstalk in TME.

Reply: We appreciate the comment. We want to perform experiments in the future.

---

## [Decision Letter · Decision Letter 1]

10 Apr 2020

Cancer cell-derived interleukin-33 decoy receptor sST2 enhances orthotopic tumor growth in a murine pancreatic cancer model

PONE-D-19-33979R1

Dear Dr. Takenaga,

We are pleased to inform you that your manuscript has been judged scientifically suitable for publication and will be formally accepted for publication once it complies with all outstanding technical requirements.

With kind regards,

Sulagna Banerjee

Academic Editor

PLOS ONE

Additional Editor Comments (optional):

Reviewers' comments:

Reviewer's Responses to Questions

**Comments to the Author**

1. If the authors have adequately addressed your comments raised in a previous round of review and you feel that this manuscript is now acceptable for publication, you may indicate that here to bypass the “Comments to the Author” section, enter your conflict of interest statement in the “Confidential to Editor” section, and submit your "Accept" recommendation.

Reviewer #1: All comments have been addressed

Reviewer #2: All comments have been addressed

2. Is the manuscript technically sound, and do the data support the conclusions?

Reviewer #1: Yes

Reviewer #2: Yes

3. Has the statistical analysis been performed appropriately and rigorously? 

Reviewer #1: Yes

Reviewer #2: Yes

4. Have the authors made all data underlying the findings in their manuscript fully available?

Reviewer #1: Yes

Reviewer #2: Yes

5. Is the manuscript presented in an intelligible fashion and written in standard English?

Reviewer #1: Yes

Reviewer #2: Yes

6. Review Comments to the Author

Reviewer #1: Author' reply for Δct plots for ST2 and sST2-We have provided a plot showing the ΔCt values in the Supporting information (S1 Fig).

Reviewer's comment- This is actually S2 Fig. Check it throughput the manuscript and make correction if any required.

Reviewer #2: (No Response)

7. PLOS authors have the option to publish the peer review history of their article (what does this mean?). If published, this will include your full peer review and any attached files.

Reviewer #1: Yes: Utpreksha Vaish

Reviewer #2: Yes: kousik kumar kesh

---

## [Editor Report · Acceptance letter]

16 Apr 2020

PONE-D-19-33979R1 

Cancer cell-derived interleukin-33 decoy receptor sST2 enhances orthotopic tumor growth in a murine pancreatic cancer model 

Dear Dr. Takenaga:

I am pleased to inform you that your manuscript has been deemed suitable for publication in PLOS ONE. Congratulations! Your manuscript is now with our production department. 

With kind regards,

on behalf of

Dr. Sulagna Banerjee 

Academic Editor

PLOS ONE